# Structural insights into the assembly and polyA signal recognition mechanism of the human CPSF complex

Marcello Clerici[1], Marco Faini[2], Ruedi Aebersold[2,3], Martin Jinek[1]*

[1]Department of Biochemistry, University of Zurich, Zurich, Switzerland; [2]Department of Biology, Institute of Molecular Systems Biology, ETH Zurich, Zurich, Switzerland; [3]Faculty of Science, University of Zurich, Zurich, Switzerland

**Abstract** 3' polyadenylation is a key step in eukaryotic mRNA biogenesis. In mammalian cells, this process is dependent on the recognition of the hexanucleotide AAUAAA motif in the pre-mRNA polyadenylation signal by the cleavage and polyadenylation specificity factor (CPSF) complex. A core CPSF complex comprising CPSF160, WDR33, CPSF30 and Fip1 is sufficient for AAUAAA motif recognition, yet the molecular interactions underpinning its assembly and mechanism of PAS recognition are not understood. Based on cross-linking-coupled mass spectrometry, crystal structure of the CPSF160-WDR33 subcomplex and biochemical assays, we define the molecular architecture of the core human CPSF complex, identifying specific domains involved in inter-subunit interactions. In addition to zinc finger domains in CPSF30, we identify using quantitative RNA-binding assays an N-terminal lysine/arginine-rich motif in WDR33 as a critical determinant of specific AAUAAA motif recognition. Together, these results shed light on the function of CPSF in mediating PAS-dependent RNA cleavage and polyadenylation.

DOI: https://doi.org/10.7554/eLife.33111.001

*For correspondence:
jinek@bioc.uzh.ch

Competing interests: The authors declare that no competing interests exist.

## Introduction

The 3'-terminal polyA tail of eukaryotic mRNAs is generated by a two-step process consisting of an initial endonucleolytic cleavage of the nascent RNA transcript followed by polyadenylation of the upstream cleavage fragment by polyA polymerase (PAP) (*Chan et al., 2011*; *Shi and Manley, 2015*). Although 3'-polyadenylation is an obligatory step in the biogenesis of non-histone mRNAs, many eukaryotic genes contain alternative polyadenylation sites that generate different protein isoforms, or mRNA isoforms with variable 3'-untranslated regions (UTRs), thereby modulating their ability to interact with microRNAs and 3'-UTR interacting factors (*Derti et al., 2012*; *Elkon et al., 2013*; *Hoque et al., 2013*; *Tian and Manley, 2017*). The selection of a specific polyadenylation site is a dynamically regulated process during cell differentiation, proliferation and development (*Sandberg et al., 2008*; *Ji et al., 2009*; *Shepard et al., 2011*; *Graber et al., 2013*), making mRNA polyadenylation a key mechanism of gene expression control.

In mammalian cells, the principal cis-acting motif within the polyadenylation signal (PAS) that defines the site of cleavage and polyadenylation is a hexanucleotide A(A/U)UAAA sequence (*Proudfoot and Brownlee, 1976*; *Chan et al., 2011*). This sequence motif (also referred to as the PAS hexamer motif or AAUAAA motif) is typically located approximately 10–30 nucleotides upstream of the endonucleolytic cleavage site, which is generally marked by the sequence CA (*Sheets et al., 1990*). The key protein factor responsible for polyA site definition is the cleavage and polyadenylation specificity factor (CPSF), a multisubunit complex that specifically recognizes the AAUAAA motif, catalyzes pre-mRNA cleavage, and recruits PAP to initiate polyadenylation at the 3' hydroxyl group of the upstream cleavage fragment. (*Takagaki et al., 1988*; *Keller et al., 1991*).

Other cis-elements in the vicinity of the AAUAAA motif that contribute to the definition of the polyA site include upstream UGUA-containing sequences (USE) (*Carswell and Alwine, 1989*; *Brackenridge and Proudfoot, 2000*) and downstream G- and GU-rich sequences (DSE) (*Gil and Proudfoot, 1984*; *McLauchlan et al., 1985*; *Gil and Proudfoot, 1987*). These elements are recognized by the cleavage factor I (CFI) and the cleavage stimulation factor (CstF) complexes, respectively (*Shi and Manley, 2015*), both of which contribute to polyA site definition and its regulation.

The initial biochemical characterization of CPSF revealed the presence of four subunits: CPSF160, CPSF100, CPSF73 and CPSF30 (*Bienroth et al., 1991*; *Murthy and Manley, 1992*). Two additional subunits integral to CPSF, Fip1 and WDR33, which are homologous to the yeast 3' processing factors Fip1p and Pfs2p, respectively, were identified at a later stage (*Kaufmann et al., 2004*; *Shi et al., 2009*). RNA cleavage is catalyzed by CPSF73, a zinc-dependent endonuclease that contains a metallo-beta-lactamase domain and a beta-CASP domain. CPSF73 has very weak enzymatic activity in isolation, implying that other CPSF subunits or 3'-end processing factors are required for efficient cleavage (*Ryan et al., 2004*; *Mandel et al., 2006*). CPSF100, despite its high structural similarity to CPSF73, is catalytically inactive and its function is hitherto unknown (*Mandel et al., 2006*). In turn, CPSF160, WDR33, CPSF30 and Fip1 have been shown to form a stable complex independently of CPSF100 and CPSF73 (*Schönemann et al., 2014*). This 'core' polyadenylation module of the CPSF complex is able to bind to an RNA containing the AAUAAA motif with nanomolar affinity ($K_D$ ~2 nM) and to promote its polyadenylation by PAP (*Schönemann et al., 2014*). Although CPSF160 had originally been identified as the subunit that mediates PAS hexamer motif recognition (*Murthy and Manley, 1995*; *Dichtl et al., 2002*), recent studies employing UV cross-linking have indicated that CPSF30 and WDR33 directly interact with the PAS hexamer instead (*Chan et al., 2014*; *Schönemann et al., 2014*). CPSF30 contains five consecutive zinc finger (ZF) domains and a C-terminal zinc knuckle domain (*Barabino et al., 1997*). The RNA binding activity of CPSF30 is primarily mediated by the second and third ZF domains (*Chan et al., 2014*). This region of CPSF30 is also targeted by the NS1 protein of the influenza virus, enabling the virus to inhibit the polyadenylation of host mRNAs encoding innate immunity factors (*Das et al., 2008*). Fip1 has been shown to regulate alternative polyadenylation (APA) in embryonic stem cells (*Lackford et al., 2014*). Consistent with these findings, interactions between Fip1 and uridine-rich RNA motifs located upstream of the PAS hexamer motif have been mapped in vivo and in vitro, suggesting that Fip1 plays an important role in modulating PAS selection (*Martin et al., 2012*; *Chan et al., 2014*; *Lackford et al., 2014*).

Despite its importance for mRNA biogenesis and the regulation of gene expression, the molecular architecture of the CPSF complex and its mechanism of PAS hexamer motif recognition are poorly understood. Here, we present structural and functional insights into the assembly of the core AAUAAA-binding module of the human CPSF complex. We show that CPSF160 and WDR33 form an heterodimer independently of the other CPSF subunits and report a 2.5 Å-resolution crystal structure of the subcomplex. We further reveal that CPSF30 orchestrates the assembly of the quaternary core CPSF module complex by bridging CPSF160-WDR33 with Fip1. Finally, we show that in addition to CPSF30 ZF domains, specific recognition of the PAS hexamer is mediated by the N-terminal region of WDR33. These results establish a framework for further mechanistic studies of the CPSF complex in eukaryotic mRNA polyadenylation.

## Results

### A core module containing the N-terminal WD40 domain of WDR33 is sufficient for PAS recognition by CPSF

A recent study demonstrated that a four-subunit CPSF core complex containing WDR33 is necessary and sufficient to support AAUAAA motif-dependent polyadenylation in vitro (*Schönemann et al., 2014*). Human WDR33 is a ~145 kDa protein composed of an N-terminal WD40 beta-propeller domain and a poorly conserved C-terminal region containing low-complexity sequences. To shed light on the assembly of the CPSF polyadenylation module, we first reconstituted by co-expression in Sf9 insect cells a minimal complex consisting of full-length human CPSF160 (CPSF160^FL), CPSF30 (CPSF30^FL) and Fip1 (Fip1^FL) proteins, and a fragment of WDR33 comprising only the N-terminal region containing the WD40 domain (WDR33^M1-K410) (*Figure 1—figure supplement 1A*). We then

tested whether the reconstituted complex is capable of specific binding to an RNA containing the PAS hexamer motif. To this end, we used a Atto[532]-labelled 16-nucleotide RNA containing the AAUAAA hexamer and tested its binding in a fluorescence polarization assay (*Figure 1A*). The RNA was bound with sub-nanomolar affinity ($K_D$ = 0.65 ± 0.09 nM), in general agreement with a previously reported value ($K_D$ ~2 nM) (*Schönemann et al., 2014*). By contrast, the affinity of the complex for a mutated version of the RNA containing a single nucleotide substitution in the hexanucleotide (AAGAAA) was reduced by more than 100-fold (*Figure 1A*). Together, these results indicate that a core module of the CPSF complex containing the WD40 domain of WDR33 is able to bind the AAUAAA motif with both high affinity and specificity, implying that the C-terminal region of WDR33 is not necessary for CPSF assembly and PAS recognition.

To dissect inter- and intra-subunit interactions within the CPSF core module, we used cross-linking coupled to mass spectrometry on a variant of the minimal CPSF complex described above, carrying a C-terminally extended version of WDR33 (WDR33[M1-G474], *Figure 1—figure supplement 1A, B*). To this end, we used the cross-linking agent disuccinimidyl suberate (DSS), which cross-links lysine residues located within approximately 35 Å of each other, and identified cross-linked peptides by mass spectrometry. We identified 54 and 99 validated inter- (*Figure 1B*) and intra-protein (*Figure 1—figure supplement 1C*) cross-linked sites, respectively (as judged by xQuest score higher than 30, corresponding to an approximate false discovery rate of 10%, *Figure 1—source data 1*). The cross-links cover most of the sequences and structured domains of the constituent subunits. We observe distinct cross-linking patterns of CPSF160 to the other three subunits. Whereas Fip1 forms cross-links along the entire CPSF160 sequence, WDR33 and CPSF30 cross-links mostly to the middle and C-terminal regions of CSPF160, respectively. Notably, lysine residues K46, K50 and K55 in the N-terminal region of WDR33 upstream of the predicted WD40 domain form a cross-linking hotspot, interacting extensively with the central part of CSPF160, all five zinc finger domains of CPSF30 and a central, highly conserved region of Fip1 (*Figure 1B*). Additionally, we identify numerous cross-links between the conserved Fip1 region and the zinc finger and the C-terminal zinc knuckle domains of CPSF30. Finally, cross-links from CSPF160 residues map exclusively to CPSF30 zinc finger domains ZF1 and ZF2 (spanning residues K35-K92). Together, these results highlight extensive inter-subunit interactions within the polyA signal-binding core module of the CPSF complex and further underscore the critical role played by WDR33 in the assembly of the CPSF complex.

## Crystal structure of the CPSF160-WDR33 heterodimer reveals the structural scaffold of the CPSF complex

To gain insights into the molecular architecture of the CPSF core complex, we reconstituted and crystallized a heterodimeric complex consisting of CPSF160[FL] (residues M1-F1443) and an N-terminal WDR33 fragment encompassing residues Q35-K410 (WDR33[Q35-K410], *Figure 1—figure supplement 1A*). The structure of the complex was solved at 2.5 Å resolution by a combination of molecular replacement and tantalum bromide (Ta$_6$Br$_{12}$) and sulphur single-wavelength anomalous dispersion (SAD), refined to an $R_{free}$ factor of 26.2% (*Table 1*). The structure revealed that CPSF160 is a multi-domain protein composed of three seven-bladed WD40 beta-propeller domains (bP1, bP2 and bP3 for beta propeller 1 to 3) and a C-terminal helical domain (CTD). The three beta-propellers form a compact tristar arrangement (*Figure 2A*), reminiscent of the domain arrangement observed in the DNA Damage Binding protein 1 (DDB1) (*Li et al., 2006*) (*Figure 2—figure supplement 1A*). The central beta-propeller domain (bP2) is located at the top of the bP1-bP3 contact region, with the CTD sealing the interaction between the three beta-propellers at their central junction (*Figure 2A*, side view). The N- and C-terminal propellers (bP1 and bP3) are oriented in a pseudosymmetric fashion and interact with each other at a ~60° angle, creating a deep cavity in between. The cavity is closed on the distal side by three elongated loops (el1-el3) comprising residues P223-T237 (el1), L300-C324 (el2) and C1044-K1069 (el3) projecting from bP1 and bP3 (*Figure 2—figure supplement 2A*). CPSF160 and DDB1 share a remarkable structural similarity despite low sequence identity (~16%), in particular for the bP1-bP3-CTD ensemble (RMSD 2.5 Å for 737 aligned Cα atoms) (*Figure 2—figure supplement 1A*). However, in contrast to DDB1, bP2 is tightly packed against the other two propeller domains and CTD in CPSF160, making extensive ionic and hydrophobic interactions (~3600 Å$^2$ of total buried area and 54 charged interactions for CPSF160 in comparison to ~1000 Å$^2$ and 10 interactions for DDB1). This tight packing suggests that bP2 is locked in a fixed position in CPSF160, whereas different crystal forms show significant conformational flexibility of

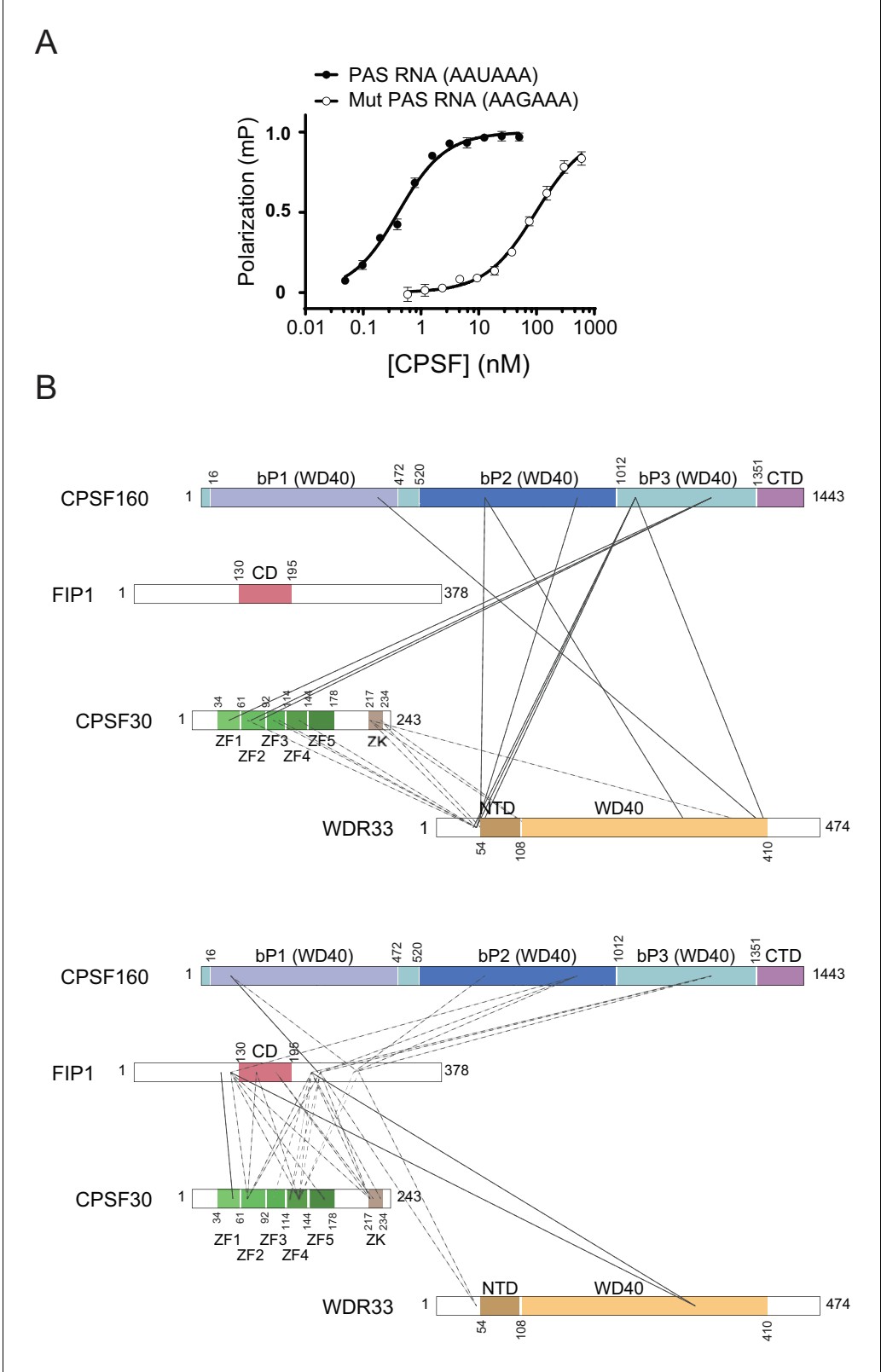

**Figure 1.** Characterization of the CPSF160-WDR33[M1-K410]-CPSF30-Fip1 complex. (**A**) Equilibrium binding of CPSF160[FL]-WDR33[M1-K410]-CPSF30[FL]-Fip1[FL] to an Atto[532]-labelled, 16-nucleotide RNA containing the PAS hexanucleotide (AAUAAA), measured by fluorescence polarization. The polarization amplitude is normalized to 1. Error bars indicate standard error of means (SEM) for five consecutive measurements of a single representative

*Figure 1 continued on next page*

*Figure 1 continued*

sample. Each binding experiment was performed in triplicate and the mean $K_D$ values ± SEM are reported in *Table 2*. (B) Inter-subunit cross-link map of the CPSF160$^{FL}$-WDR33$^{M1-G474}$-CPSF30$^{FL}$-Fip1$^{FL}$ complex. Observed inter-molecular cross-links between CPSF160$^{FL}$, WDR33$^{M1-G474}$ and CPSF30$^{FL}$ (upper panel) and between Fip$^{FL}$ and the other three CPSF subunits CPSF160$^{FL}$, WDR33$^{M1-G474}$ and CPSF30$^{FL}$ (lower panel) are represented as dashed lines. A list of the cross-linked peptides identified by mass spectrometry is reported in *Figure 1—source data 1*. Proteins are indicated as schematic diagrams (not to scale) with featured domains highlighted in color. (CD, Fip1 conserved domain; ZF1-5, CPSF30 zinc finger domains 1 to 5; ZK, CPSF30 zinc knuckle domain; CTD, CPSF160 C-terminal domain; bP1-3, CPSF160 beta-propeller domains 1 to 3; NTD, WDR33 N-terminal domain. CPSF160: Uniprot Q10570; WDR33: Uniprot Q9C0J8-1; CPSF30: Uniprot O95639-3; Fip1: Uniprot Q6UN15-4).
DOI: https://doi.org/10.7554/eLife.33111.002

The following source data and figure supplement are available for figure 1:

**Source data 1.** Table of cross-linking MS identification.
DOI: https://doi.org/10.7554/eLife.33111.004
**Figure supplement 1.** Reconstitution of the CPSF core module and intra-subunit cross-links.
DOI: https://doi.org/10.7554/eLife.33111.003

---

bP2 in DDB1 (*Li et al., 2006*) (*Figure 2—figure supplement 1A*). As in DDB1, the three beta-propeller domains of CPSF160 do not follow a linear tandem topology; instead, three discontinuous regions of the CPSF160 polypeptide sequence contribute to bP3 (*Figure 1B*).

WDR33 also forms a seven-bladed beta-propeller domain (corresponding to residues T108-T403) and additionally contains a ~50 residue N-terminal domain (NTD, residues R54-T107) protruding away from the beta-propeller (*Figure 1A* and *Figure 2B*). The NTD has a unique fold with no secondary structure features except for an N-terminal alpha-helix. The NTD is further supported by a short loop (residues R404-K410) extending C-terminally from the last beta-strand of the WD40 propeller. Upstream of the NTD, residues Q35-N53, which were included in the crystallized protein construct, are disordered in the crystal structure and could not be modeled (*Figure 2B*). The WDR33 NTD is completely buried in the cavity formed by CPSF160 bP1 and bP3 (*Figure 2C* and *Figure 2—figure supplement 2B*), with CPSF160 loops el1, el2 and el3 locking the NTD in place (*Figure 2—figure supplement 2A*). The CPSF160-WDR33 complex is further stabilized by contacts between WDR33 beta-propeller domain and CPSF160 bP1 and bP3. The CPSF160-WDR33 interaction is strikingly reminiscent of the architecture of the DDB1-DDB2 DNA repair complex (*Scrima et al., 2008*). Superposition of CPSF160 and DDB1 results in almost perfect overlap of the N-terminal alpha-helices of WDR33 and DDB2. However, due to local differences in the N-terminal domains WDR33 and DDB2, the respective beta-propeller domains of the two proteins are shifted by ~10 Å relative to each other (*Figure 2—figure supplement 1B,C*). Overall, CPSF160 and WDR33 establish an extensive network of interactions encompassing 45 hydrogen bonds and 19 salt bridges, with a total buried surface of ~6900 Å$^2$, equally distributed over the two proteins. These structural observations are thus consistent with CPSF160 and WDR33 forming a tight heterodimeric subcomplex within the polyA signal-binding module of the human CPSF complex.

The molecular architecture of the CPSF160-WDR33 heterodimer is in good agreement with our cross-linking mass spectrometry data (*Figure 1B* and *Figure 1—source data 1*), despite many of the cross-links originating from lysine residues within disordered regions of CPSF160 (internal loops) and WDR33 (N-terminal region). Among the cross-links that can be mapped on the atomic model of the CPSF160-WDR33 heterodimer, 12 are consistent with the conventional Euclidean distance of 35 Å. Notably, of the six cross-links with distance violations, five originate from the same CPSF160 residue (see Materials and methods). Four cross-links between CPSF160 K1055 located in loop el3 and WDR33 residues K46, K50, K55 and K410 are consistent with the CPSF160 beta-propeller three being in proximity of the N-terminal region of WDR33 (*Figure 1B* and *Figure 2A*).

## Molecular topology of the core CPSF complex

In light of the crystal structure of the CPSF160-WDR33 heterodimer, we sought to investigate the physical interactions linking CPSF30 and Fip1 to CPSF160-WDR33 within the AAUAAA-binding core module of CPSF. To this end, we co-expressed CPSF160$^{FL}$ and WDR33$^{M1-K410}$ together with a series of N-terminally green fluorescent protein (GFP)-tagged constructs of CPSF30 in Sf9 cells, making

**Table 1.** Data collection and refinement statistics.

| Dataset | CPSF160-WDR33 | | |
| --- | --- | --- | --- |
| | Native | Sulfur SAD | Ta$_6$Br$_{12}$ SAD |
| X-ray source | SLS X06DA (PXIII) | SLS X06DA (PXIII) | SLS X06DA (PXIII) |
| Space group | $P_1$ | $P_1$ | $P_1$ |
| Cell dimensions | | | |
| $a, b, c$ (Å) | 67.91 77.40 104.02 | 67.88 77.58 104.14 | 67.52 76.79 104.02 |
| $\alpha, \beta, \gamma$ (°) | 87.56 76.41 67.00 | 87.39 76.60 66.76 | 87.36 76.72 66.30 |
| Wavelength (Å) | 1.0000 | 2.0733 | 1.2548 |
| Resolution (Å)* | 47.27–2.50 (2.59–2.50) | 44.25–3.00 (3.11–3.00) | 47.26–3.60 (3.73–3.60) |
| $R_{merge}$* | 0.090 (0.773) | 0.211 (2.934) | 0.140 (0.669) |
| CC1/2* | 0.999 (0.834) | 0.999 (0.847) | 0.998 (0.926) |
| I/σI* | 18.3 (2.5) | 29.3 (2.5) | 20.6 (4.6) |
| Observations* | 488656 (34882) | 2456831 (165849) | 300048 (29753) |
| Unique reflections* | 65410 (6519) | 37770 (3688) | 21497 (2126) |
| Multiplicity* | 7.5 (5.4) | 65.0 (45.0) | 14.0 (14.0) |
| Completeness (%)* | 100.0 (100.0) | 100.0 (96.9) | 99.9 (99.6) |
| Refinement | | | |
| Resolution (Å) | 47.27–2.50 | | |
| No. reflections | 65395 (6517) | | |
| $R_{work}$/$R_{free}$ | 0.228/0.263 | | |
| No. atoms | | | |
| Protein | 12162 | | |
| Water | 102 | | |
| B-factors | | | |
| mean | 69.35 | | |
| Protein | 69.52 | | |
| Water | 49.03 | | |
| R.m.s. deviations | | | |
| Bond lengths (Å) | 0.002 | | |
| Bond angles (°) | 0.56 | | |
| Ramachandran plot | | | |
| % favored | 94.6 | | |
| % allowed | 5.4 | | |
| % outliers | 0.0 | | |

DOI: https://doi.org/10.7554/eLife.33111.008

use of the GFP tag for direct in-gel detection (*Figure 3A*, *Figure 1—figure supplement 1A*). In tandem affinity purifications utilizing the His$_6$-(StrepII)$_2$ epitope tag on WDR33, we initially observed efficient co-precipitation of full-length CPSF30 (CPSF30$^{FL}$, *Figure 3A*, lane 1), indicating that CPSF30 is able to interact with CPSF160$^{FL}$-WDR33$^{M1-K410}$ independently of Fip1. Next, we examined a series of CPSF30 construct with different C-terminal deletions (*Figure 3A*, lanes 2–4). All tested CPSF30 constructs retained binding to CPSF160$^{FL}$-WDR33$^{M1-K410}$, indicating that the N-terminal ~60 residues of CPSF30 that include ZF1 are sufficient for the interaction with CPSF160-WDR33 (*Figure 3A*, lane 4). Moreover, deletion of the same residues indeed impaired CPSF30 binding to CPSF160$^{FL}$-WDR33$^{M1-K410}$ (CPSF30$^{ZF2-5}$, *Figure 3A*, lane 5). A similar result was obtained when the N-terminal 34 residues of CPSF30 (upstream of ZF1) were deleted (CPSF30$^{ZF1-5\Delta N}$, *Figure 3A*, lane 6), but these residues alone were not sufficient to mediate binding to CPSF160$^{FL}$-WDR33$^{M1-K410}$ (CPSF30$^{N}$,

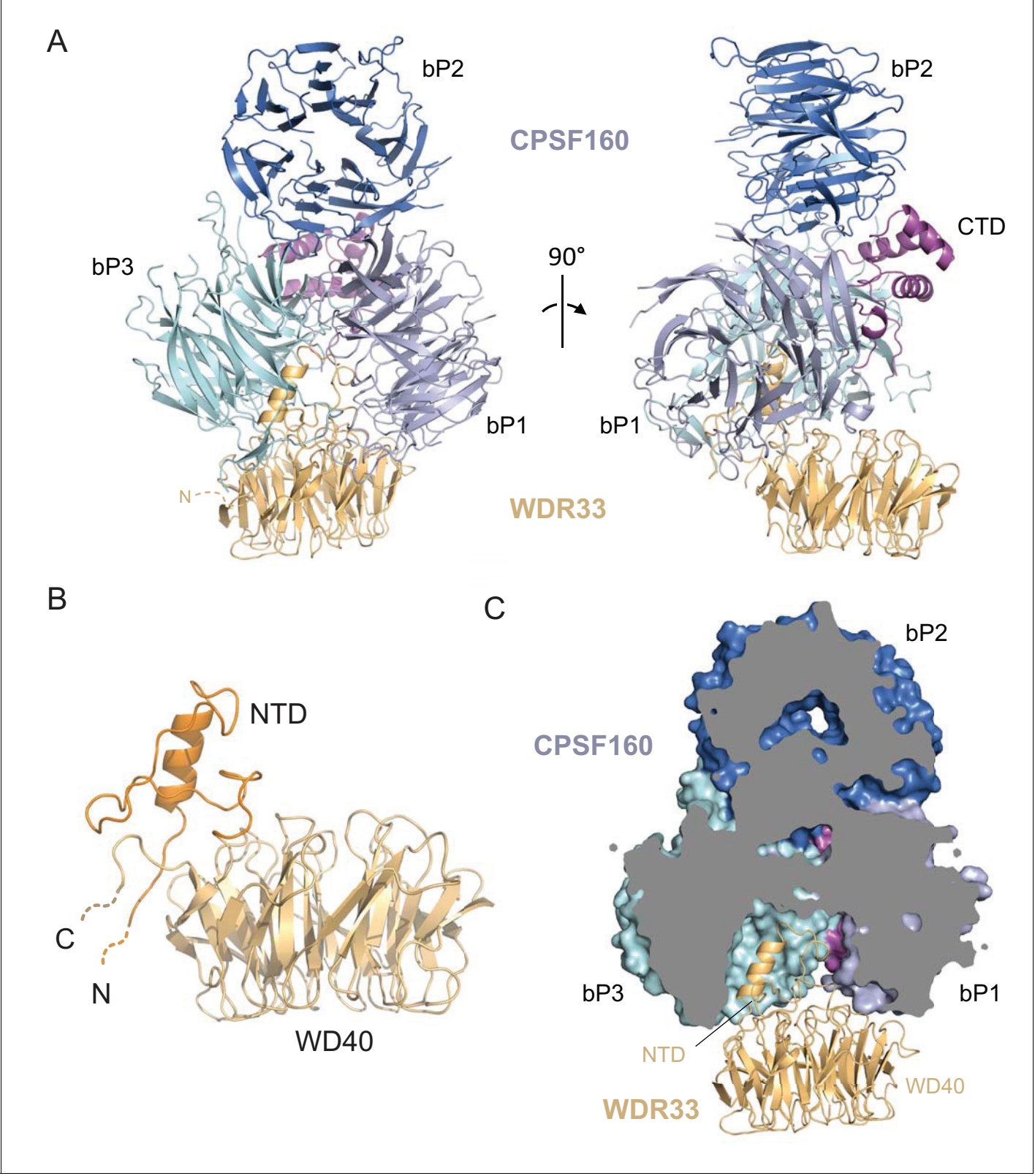

**Figure 2.** Crystal structure of the CPSF160-WDR33 complex. (**A**) Cartoon representation of the overall structure of the CPSF160$^{FL}$-WDR33$^{Q35-K410}$ complex. (**B**) Cartoon representation of the isolated WDR33 WD40 (light orange) and NTD (dark orange) domains. The N and C-termini are indicated as dashed lines. (**C**) CPSF160 beta-propeller domains bP1 and bP3 form a deep cavity that binds the WDR33 N-terminal domain (NTD). A vertical cross-section through the CPSF160 propeller domains (represented as surface) is shown. WDR33 is represented as cartoon.

*Figure 2 continued on next page*

*Figure 2 continued*

DOI: https://doi.org/10.7554/eLife.33111.005

The following figure supplements are available for figure 2:

**Figure supplement 1.** The CPSF160-WDR33 heterodimer resembles the DDB1-DDB2 complex.

DOI: https://doi.org/10.7554/eLife.33111.006

**Figure supplement 2.** WDR33 is buried within the cavity formed by CPSF160 bP1 and bP3 domains.

DOI: https://doi.org/10.7554/eLife.33111.007

*Figure 3A*, lane 7). Together, these results indicate that the N-terminal region of CPSF30 up to and including ZF1 is necessary and sufficient to mediate the interaction between CPSF30 and the CPSF160-WDR33 heterodimer.

To probe the interactions of Fip1 with the other subunits, we co-expressed in Sf9 cells GFP-tagged Fip1$^{FL}$ (*Figure 1—figure supplement 1A*) with CPSF160$^{FL}$ and His$_6$-(StrepII)$_2$-tagged WDR33$^{M1-K410}$ in the presence or absence of CPSF30$^{FL}$ and performed a pull-down experiment. In the presence of CPSF30$^{FL}$, Fip1$^{FL}$ was efficiently co-precipitated together with the rest of the

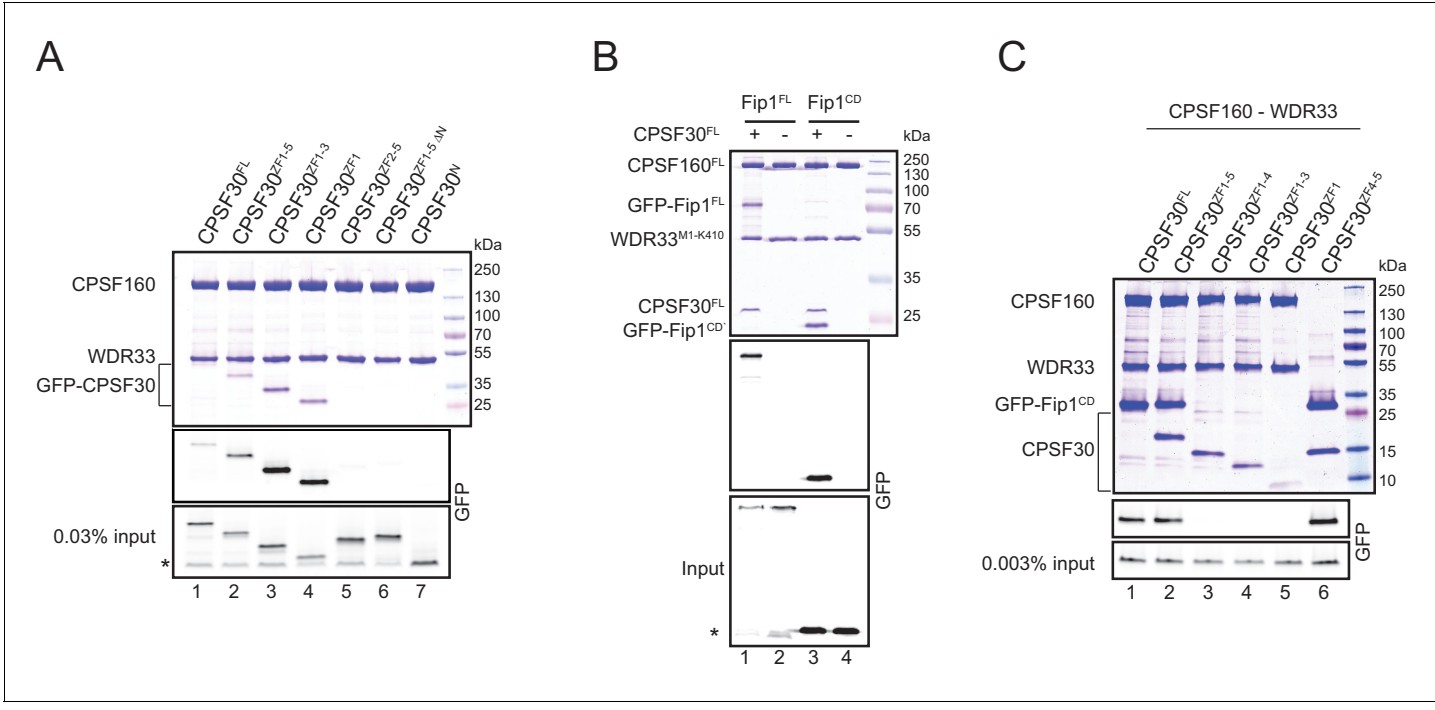

**Figure 3.** Molecular topology of the CPSF complex. (**A**) CPSF30 ZF1 is necessary and sufficient for CPSF160$^{FL}$-WDR33$^{M1-K410}$ binding. Pull-down assay of GFP-labeled CPSF30 constructs interacting with CPSF160$^{FL}$-WDR33$^{M1-K410}$. Input and bound proteins were analyzed on a 4–12% gradient SDS-PAGE and visualized by Coomassie staining (upper panel) and by in-gel GFP fluorescence (middle and lower panels). The asterisk indicates free GFP present in the input lysate. (**B**) Fip1 is tethered to the CPSF complex by its short conserved domain (Fip1$^{CD}$). Pull-down assay of GFP-labeled Fip1$^{FL}$ and Fip1$^{CD}$ interacting with the CPSF160$^{FL}$-WDR33$^{M1-K410}$-CPSF30$^{FL}$ complex. Input and bound proteins were analyzed on a 4–12% gradient SDS-PAGE and visualized by Coomassie staining (upper panel) and by in-gel GFP fluorescence (middle and lower panels). The asterisk indicates free GFP present in the input lysate. (**C**) CPSF30 zinc finger domains ZF4 and ZF5 are necessary and sufficient to mediate the interaction with Fip1. Pull-down assay of GFP-labeled Fip1$^{CD}$ interacting with CPSF160$^{FL}$-WDR33$^{M1-K410}$-CPSF30. Input and bound proteins were analyzed on a 4–12% gradient SDS-PAGE and visualized by Coomassie staining (upper panel) and by in-gel GFP fluorescence (middle and lower panels). In lane 6, His$_6$-(StrepII)$_2$-CPSF30$^{ZF4-5}$ was used for the precipitation of GFP-Fip1$^{CD}$, in the absence of WDR33$^{M1-K410}$ and CPSF160$^{FL}$.

DOI: https://doi.org/10.7554/eLife.33111.009

The following figure supplements are available for figure 3:

**Figure supplement 1.** The conserved middle domain of Fip1 (Fip1$^{CD}$) interacts with CPSF30 zinc finger domains ZF4 and ZF5.

DOI: https://doi.org/10.7554/eLife.33111.010

**Figure supplement 2.** Mapping of CPSF30 and Fip1 cross-links onto the structure of CPSF160-WDR33.

DOI: https://doi.org/10.7554/eLife.33111.011

complex (*Figure 3B*, lane 1). In contrast, Fip1[FL] did not interact with CPSF160[FL] and WDR33 [M1-K410] in the absence of CPSF30[FL] expression (lane 2). Human Fip1 contains a central sequence motif that is highly conserved across organisms, including yeast (*Figure 3—figure supplement 1A*). A ~7 kDa fragment encompassing this conserved motif, spanning residues G130-K195, (henceforth termed the Fip1 conserved domain, Fip1[CD], *Figure 1—figure supplement 1A*), retained the ability to interact with the CPSF30[FL]-CPSF160[FL]-WDR33 [M1-K410] complex (*Figure 3B*, lane 3). As for Fip1[FL], binding was dependent on the presence of CPSF30[FL] (*Figure 3B*, lanes 3–4), indicating that a direct physical interaction between CPSF30 and Fip1[CD] is required to recruit Fip1 to the CPSF core module. Altogether, these results thus indicate that Fip1 is tethered to the CPSF complex via CPSF30, although additional weaker interactions with CPSF160 and WDR33 cannot be excluded based on our cross-linking and mass spectrometry data (*Figure 1B*). Moreover, the conservation of the Fip1[CD] motif suggests that this mode of interaction is also conserved in the yeast CPF complex, the functional equivalent of metazoan CPSF.

To delineate specific domains in CPSF30 responsible for Fip1 interaction, we conducted further co-precipitation experiments using GFP-tagged Fip1[CD] and C-terminally truncated CPSF30 variants. Whereas both CPSF30[FL] and a construct comprising all five ZF domains (CPSF30[ZF1-5]) exhibited binding to Fip1 (*Figure 3C*, lanes 1–2), constructs lacking just ZF5 (CPSF30[ZF1-4], *Figure 3C*, lane 3) or additional ZF domains (CPSF30[ZF1-3] and CPSF30[ZF1], *Figure 3C*, lanes 4–5) showed impaired binding, indicating that ZF5 is necessary for Fip1 interaction. In the absence of CPSF160 and WDR33, a His$_6$-(StrepII)$_2$–tagged fragment of CPSF30 comprising the ZF4 and ZF5 domains (CPSF30[ZF4-5]) was able to interact with Fip1[CD] (*Figure 3C*, lane 6). To corroborate this finding, we performed co-precipitation assays using bacterially expressed recombinant proteins. A maltose-binding protein (MBP)-tagged CPSF30 construct containing zinc finger domains ZF1-5 (CPSF30[ZF1-5]) was efficiently co-precipitated by glutathione S-transferase (GST-) fused Fip1[CD] (*Figure 3—figure supplement 1B*). Interestingly, MBP-tagged CPSF30 construct comprising domains ZF1-4 (CPSF30[ZF1-4]) retained some residual binding to Fip1[CD], whereas a construct comprising ZF1-3 (CPSF30[ZF1-3]) did not, suggesting that ZF4 also contributes to the CPSF30-Fip1 interaction in addition to ZF5.

Taken together, these results indicate that: (i) CPSF30 binds to CPSF160-WDR33 independently of Fip1 and that its N-terminal ~60 residues (including ZF1) are necessary and sufficient for this interaction; (ii) Fip1 is tethered to the CPSF complex by its short ~7 kDa conserved domain (Fip1[CD]); and (iii) the zinc finger domains ZF4 and ZF5 of CPSF30 are necessary and sufficient to mediate the interaction with Fip1. Furthermore, these conclusions are consistent with the cross-linking data between each subunit (*Figure 1B*), as mapping the molecular cross-links of CPSF30 and Fip1 onto the molecular surface of the CPSF160-WDR33 heterodimer suggests that CPSF30 binds at, or close to, the CPSF160-WDR33 molecular interface in proximity to the N-terminus of WDR33 (*Figure 3—figure supplement 2*).

## AAUAAA motif recognition is mediated by CPSF30 ZF2-3 domains and an N-terminal motif in WDR33

As cross-linking and immunoprecipitation studies previously implicated human CPSF30 and WDR33 in direct interactions with the AAUAAA signal (*Chan et al., 2014*; *Schönemann et al., 2014*), we sought to investigate the structural requirements for PAS hexamer motif binding by the CPSF complex and dissect the contributions of its individual subunits to AAUAAA motif recognition. To this end, we quantified the binding of an AAUAAA-containing RNA by the CPSF core module and variants thereof in a fluorescence polarization assay. The core module containing CPSF30[FL] and Fip1[FL] bound the AAUAAA RNA with subnanomolar affinity (*Figure 1A*, *Figure 4A*). The affinity was sustained and even increased with a complex containing Fip1[CD] and CPSF30 that lacked the C-terminal ZK domain (CPSF30[ZF1-5], *Figure 4* and *Table 2*). A complex lacking Fip1 and containing only CPSF30 domains ZF1-3 (CPSF30[ZF1-3]) also bound the AAUAAA RNA with extremely high affinity, with our assay setting an upper boundary of ~100 pM for the equilibrium dissociation constant. Although the observed increase in affinity upon removal of CPSF30 domains ZF4-5 and Fip1 is presently unclear, it is conceivable that CPSF30 ZF4-5 and/or Fip1 play an autoinhibitory role within CPSF by sterically hindering the accessibility of CPSF30 ZF2-3 domains to the AAUAAA motif.

In contrast, deletion of the CPSF30 ZF3 domain (CPSF30[ZF1-2]) resulted in a dramatic (>2000 fold) reduction in AAUAAA RNA binding, indicating that the ZF3 domain is essential for the recognition of the PAS hexamer motif and that the presence of the ZF2 domain alone is not sufficient (*Figure 4*

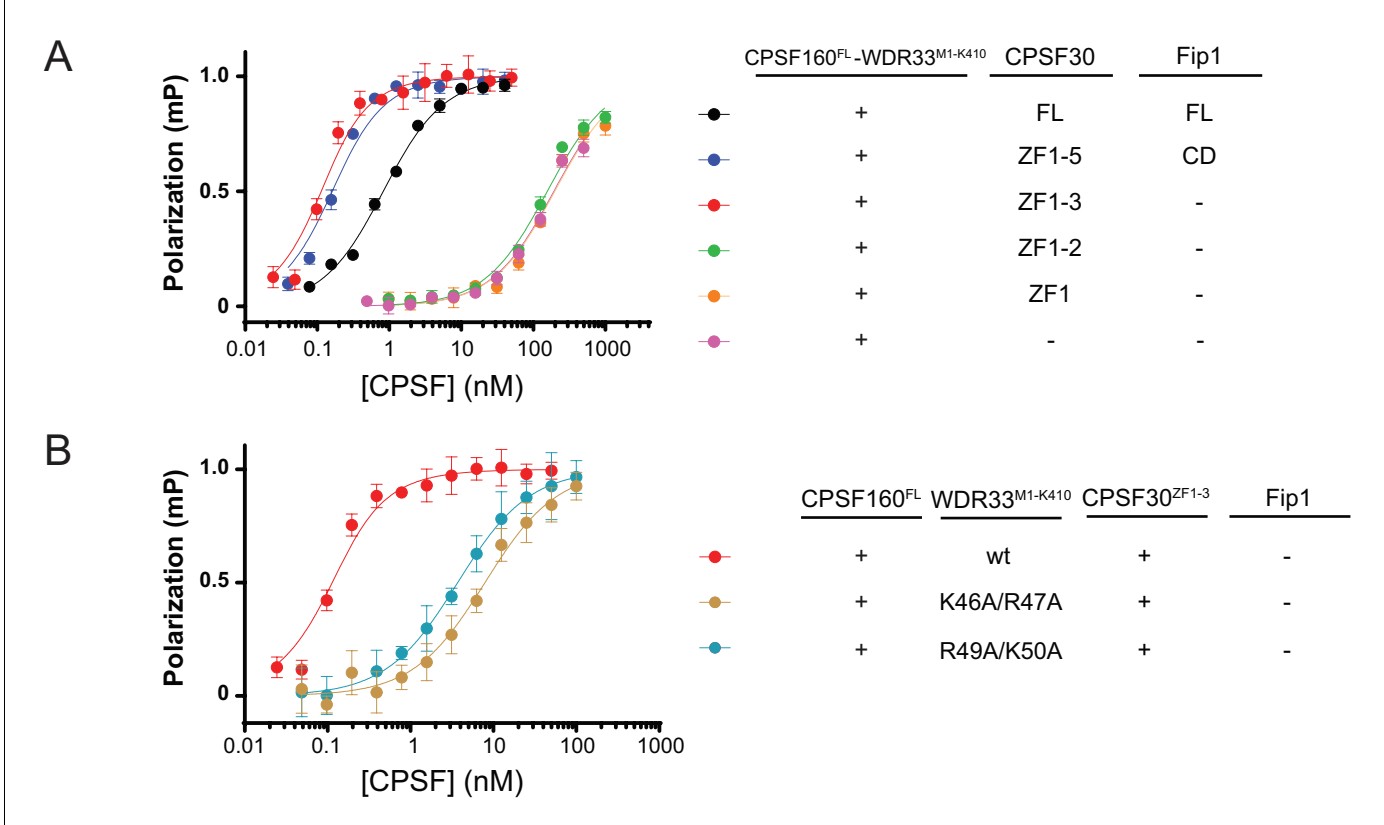

**Figure 4.** CPSF30 ZF3 domain and WDR33 N-terminal motif are principal determinants of AAUAAA motif. Equilibrium binding measured by fluorescence polarization of CPSF complex variants containing different CPSF30 constructs (**A**) and WDR33$^{M1-K410}$ mutants (**B**) to an Atto$^{532}$-labelled 16-nucleotide RNA containing the PAS hexamer (AAUAAA). The polarization amplitude is normalized to 1. Error bars indicate standard error of means (SEM) for five consecutive measurements of the same sample. Error bars indicate standard error of means (SEM) for five consecutive measurements of a single representative sample. Each binding experiment was performed in triplicate and the mean $K_D$ values ± SEM are reported in **Table 2**.

DOI: https://doi.org/10.7554/eLife.33111.012

The following figure supplement is available for figure 4:

**Figure supplement 1.** Quantification of AAUAAA motif binding by CPSF complexes.

DOI: https://doi.org/10.7554/eLife.33111.013

and **Table 2**). This is in good agreement with previous experiments demonstrating that both the ZF2 and ZF3 domains are required for interactions with the AAUAAA motif (**Chan et al., 2014**). Surprisingly, however, no further reduction in AAUAAA RNA binding was observed when CPSF30 was further truncated to remove the ZF2 domain (CPSF30$^{ZF1}$), or omitted from the core module altogether, indicating that in the absence of the ZF3 domain, the ZF2 domain of CPSF30 does not contribute to AAUAAA binding. Binding was significantly decreased for all CPSF complex variants when a mutant RNA containing a single-base substitution in the PAS hexamer motif (AAGAAA) was used (**Figure 4— figure supplement 1A**), indicating that some specificity for the AAUAAA RNA is retained also in the absence of CPSF30. Given that WDR33, as opposed to CPSF160, has been implicated in AAUAAA recognition (**Chan et al., 2014**; **Schönemann et al., 2014**), this suggests that WDR33 indeed contributes to specific recognition of the PAS hexanucleotide.

We next sought to identify specific features in WDR33 involved directly in AAUAAA motif recognition. Our cross-linking and mass spectrometry analysis indicated that an N-terminal motif located immediately upstream of the NTD in WDR33 is a cross-linking hotspot. This region, which is not resolved in the structure of the CPSF160-WDR33 subcomplex, is the only part of WDR33 that cross-links to CPSF30 ZF2 and ZF3 domains, pointing to the spatial proximity of the two regions in the CPSF subunits (**Figure 1B**). Furthermore, the sequence of the N-terminal motif is characterized by a highly conserved pattern of positively charged residues (K46-K55) that might be involved in RNA

**Table 2.** Equilibrium dissociation constants of CPSF variants binding to wild-type or mutated PAS hexamer RNA.

The equilibrium dissociation constants were determined using a fluorescence polarization binding assay, as shown in *Figures 1A* and *4A,B* and *Figure 4—figure supplement 1A*. The values reported represent mean ±SEM of three independent measurements. For measurements denoted n.m. (not measurable), the $K_D$ was above the range measurable by the assay (>>1 μM).

| CPSF variant | $K_D$ AAUAAA RNA | $K_D$ AAGAAA RNA |
|---|---|---|
| CPSF160[FL]-WDR33[M1-K410]-CPSF30[FL]/Fip1[FL] | 0.65 ± 0.90 nM | 120 ± 23 nM |
| CPSF160[FL]-WDR33[M1-K410]-CPSF30[ZF1-5]/Fip1[CD] | 0.11 ± 0.03 nM | 70 ± 6 nM |
| CPSF160[FL]-WDR33[M1-K410]-CPSF30[ZF1-3] | <0.1 nM | 43 ± 2 nM |
| CPSF160[FL]-WDR33[M1-K410]-CPSF30[ZF1-2] | >200 nM | n.m. |
| CPSF160[FL]-WDR33[M1-K410]-CPSF30[ZF1] | >200 nM | n.m. |
| CPSF160[FL]-WDR33[M1-K410] | >200 nM | n.m. |
| CPSF160[FL]-WDR33[M1-K410](K46A-R47A)-CPSF30[ZF1-3] | 7.78 ± 0.78 nM | >200 nM |
| CPSF160[FL]-WDR33[M1-K410](R49A-K50A)-CPSF30[ZF1-3] | 2.42 ± 0.66 nM | >500 nM |

DOI: https://doi.org/10.7554/eLife.33111.014

binding by ionic or cation-π interactions (*Figure 4—figure supplement 1B*). Based on these observations, we designed two WDR33[M1-K410] constructs in which specific lysine and arginine residues in the N-terminal motif were substituted with alanine (WDR33[M1-K410] K46A/R47A and R49A/K50A). These mutant proteins were still able to support the assembly of the CPSF160[FL]-WDR33[M1-K410]-CPSF30[ZF1-3] complex (*Figure 4—figure supplement 1C*), indicating that the N-terminal motif is not required for the assembly of the core CPSF module. Next, we tested binding of the resulting complexes to AAUAAA RNA in a fluorescence polarization assay. Both sets of mutations in WDR33 resulted in substantial reduction in affinity, with K46A/R47A binding with an equilibrium dissociation constant of 7.78 ± 0.78 nM (a ~70 fold reduction compared to wild-type WDR33[M1-K410]) and R49A/K50A with a $K_D$ of 2.42 ± 0.66 nM (~25 fold reduction) (*Figure 4B* and *Table 2*). Notably, binding assays with the mutant RNA containing a single-base substitution in the PAS hexamer motif (AAGAAA) revealed that the K46A/R47A substitution in WDR33 results in a ~5 fold reduction in affinity compared to wild-type WDR33[M1-K410] (200 nM versus 43 nM, *Table 2*), whereas the R49A/K50A substitution leads to ~12 fold reduction in affinity (500 nM versus 43 nM, *Table 2*). These results indicate that the positively charged lysine and arginine residues in the N-terminal region of WDR33 are involved in AAUAAA motif recognition and contribute both to the affinity and specificity of AAUAAA binding. Taken together, these experiments identify the ZF2 and ZF3 domains of CPSF30 and the N-terminal motif of WDR33 as the critical determinants of AAUAAA recognition within the CPSF complex.

## Discussion

A four-subunit core module of the mammalian CPSF complex has previously been shown to be sufficient for the recognition of the AAUAAA hexamer motif of the polyadenylation signal via its CPSF30 and WDR33 subunits (*Schönemann et al., 2014*). In this work, we sought to shed light on the structural architecture of the core module of human CPSF and its molecular mechanism of AAUAAA motif recognition. Using cross-linking and mass spectrometry analysis, we define interactions within the core module, revealing that WDR33 is a key component of the complex and identifying its N-terminal region as an interaction hotspot in the absence of bound RNA, positioned in close proximity to the remaining three subunits. The crystal structure of the CPSF160[FL]-WDR33[Q35-K410] subcomplex reveals extensive interaction between the two proteins resulting from shape complementarity of the N-terminal domain of WDR33 and a deep cavity organized by the beta-propeller domains and extended loops in CPSF160. The CPSF160-WDR33 subcomplex thus constitutes the structural scaffold of the core polyadenylation module of the CPSF complex and provides an interaction platform for the other CPSF subunits, including CPSF100 and CPSF73. Surprisingly, the architecture of the CPSF160-WDR33[Q35-K410] heterodimer, consisting of four WD40 beta-propeller domains, is highly reminiscent of the DDB1-DDB2 complex involved in the detection and repair of UV-induced DNA

damage, suggesting distant evolutionary relationships between the two molecular machineries. A recent cryo-EM structure of the yeast Cft1-Pfs2-Yth1-Fip1 complex, which is orthologous and functionally equivalent to the human CPSF160-WDR33-CPSF30-Fip1 complex and constitutes the polyadenylation module of the yeast CPF complex, revealed striking similarities to the crystal structure of the CPSF160-WDR33 heterodimer (*Casañal et al., 2017*). The yeast Cft1-Pfs2 subcomplex superimposes with the human CPSF160-WDR33 heterodimer with RMSDs of 2.56 Å for CPSF160 and Cft1 and 1.55 Å for WDR33 and Pfs2, indicating that the core CPSF scaffold is highly evolutionarily conserved. Together, these structures reveal a high degree of structural homology between the yeast CPF and mammalian CPSF complexes. Overall, the structure of the yeast complex is supportive of the data presented in our study, as further discussed below.

Using co-expression and co-precipitation experiments, we dissect the individual inter-subunit interactions underpinning the assembly of the core CPSF polyadenylation module. In addition to its critical role in AAUAAA binding, CPSF30 functions in linking the CPSF160-WDR33 heterodimer and Fip1. While the N-terminal ZF1 domain of CPSF30 is required for the interaction with CPSF160-WDR33, the zinc finger domains ZF4 and ZF5 are involved in the recruitment of Fip1. This is good agreement with previous studies of the yeast CPSF30 ortholog Yth1 showing that deletion of the fifth zinc finger domain impairs interactions with Fip1p and Pap1p (the yeast orthologs of Fip1 and PAP) and causes a polyadenylation defect in vitro (*Barabino et al., 2000*). Our interaction data are consistent with the yeast Cft1-Pfs2-Yth1-Fip1 complex structure, in which the N-terminal region of Yth1 up to and including ZF1 mediates the interaction with Cft1-Pfs2 (CPSF160-WDR33) while the C-terminal ZF4 and ZF5 domains, together with Fip1, are structurally disordered (*Casañal et al., 2017*). Notably, Yth1 ZF1 and ZF2 domains are located in the cleft formed by Cft1 and Pfs2, in good agreement with our cross-linking data.

Using RNA-binding assays with complexes containing truncated CPSF30 proteins, we show that the CPSF30 ZF2 and ZF3 domains play key roles in the specific recognition of the AAUAAA motif. ZF3 additionally appears to play a structural role in the positioning of ZF2, since ZF2 does not appear to be able to support RNA binding in the absence of ZF3. These results are in agreement with experimental results showing that deletion of the ZF2 domain in the context of full-length human CPSF30 abrogates AAUAAA RNA interactions in vitro, while point mutations in yeast Yth1p that likely result in the unfolding of the second zinc finger domain abrogate mRNA polyadenylation in vivo and in cell extracts (*Barabino et al., 2000*; *Chan et al., 2014*). The CPSF30 zinc knuckle domain is not essential for the assembly of the core CPSF polyadenylation module, which is consistent with the absence of this domain in yeast Yth1. However, deletion of the ZK domain in human CPSF30 reduces the efficiency of its cross-linking to AAUAAA-containing RNA in vitro (*Chan et al., 2014*), suggesting that the ZK domain also contributes to RNA binding. However, the precise role of the ZK domain remains to be investigated.

Our co-precipitation and RNA binding assays reveal that Fip1 is a peripheral subunit that is dispensable for specific recognition of the AAUAAA motif. The central conserved domain of Fip1 (Fip1$^{CD}$) interacts with the ZF4 and ZF5 domains of CPSF30, as observed for the yeast orthologs (*Barabino et al., 2000*). Given the conservation of the Fip1$^{CD}$ in yeast and protozoa, this interaction mode is conserved across all eukaryotes, which enables the cleavage and polyadenylation complexes to recruit, via Fip1, other polyadenylation factors including polyA polymerase and CStF (or the related factor CFI in yeast). Given that Fip1 has intrinsic RNA-binding activity (*Kaufmann et al., 2004*), the interaction also likely enables mammalian Fip1 to modulate polyA site definition, which might be critical for the recently uncovered role of Fip1 in embryonic stem cell renewal and somatic cell reprogramming (*Lackford et al., 2014*).

Although CPSF160 was originally assumed to be the CPSF subunit involved in AAUAAA RNA binding, recent biochemical studies have provided compelling evidence that AAUAAA recognition is mediated by WDR33 instead (*Chan et al., 2014*; *Schönemann et al., 2014*). WDR33 can be cross-linked to AAUAAA-containing RNA in vitro and transcriptome-wide photoactivatable ribonucleoside-enhanced cross-linking and immunoprecipitation (PAR-CLIP) analysis reveals that WDR33 binds near the AAUAAA motif in vivo with high specificity (*Chan et al., 2014*; *Schönemann et al., 2014*). Using in vitro RNA-binding assays, we show that the CPSF160-WDR33 subcomplex retains substantial affinity and specificity for the AAUAAA motif in the absence of CPSF30, indicating that WDR33 indeed contributes to specific recognition of the AAUAAA hexanucleotide. Upstream of the NTD, human WDR33 contains a lysine/arginine-rich motif that is highly conserved in higher eukaryotes but

not in the yeast orthologue Pfs2p (*Figure 4—figure supplement 1B*). In the absence of bound AAUAAA RNA, the N-terminal motif is a cross-linking hotspot located in physical proximity to the CPSF30 ZF3 domain, suggesting that the motif is involved in AAUAAA recognition. Consistent with these observations, mutations of specific lysine or arginine residues in the motif impair AAUAAA motif binding by the core CPSF module in vitro. Crucially, these mutations do not perturb module assembly, suggesting that the WDR33 N-terminal motif is not involved in CPSF30 recruitment and instead mediates direct interactions with the AAUAAA motif. Unlike in mammals, where polyA site definition is strictly dependent on the A(A/U)UAAA hexanucleotide, the yeast cleavage and polyadenylation machinery recognizes a degenerate, A-rich sequences known as the positioning element (*Guo and Sherman, 1995*; *Chan et al., 2011*). It is tempting to speculate that the difference in the N-termini of human WDR33 and yeast Pfs2p might account for the differences in polyA site definition by the yeast and mammalian polyadenylation machineries.

Based on our structural observations and interaction analysis, we conclude that the CPSF160-WDR33 heterodimer functions as an interaction platform, recruiting CPSF30 and indirectly Fip1, via the ZF1 domain in CPSF30, and interacting with additional subunits of the CPSF complex including CPSF100 and CPSF73 (*Figure 5*). AAUAAA motif binding occurs at the subunit interface of CPSF160 and WDR33 in close proximity to the WDR33 N-terminus, where the N-terminal lysine/arginine-rich motif of WDR33 and the ZF2 and ZF3 domains of CPSF30 act synergistically to recognize the AAUAAA motif in a sequence-specific manner. This enables the CPSF complex to bind the AAUAAA motif with subnanomolar affinity and remarkable specificity. While this manuscript was in

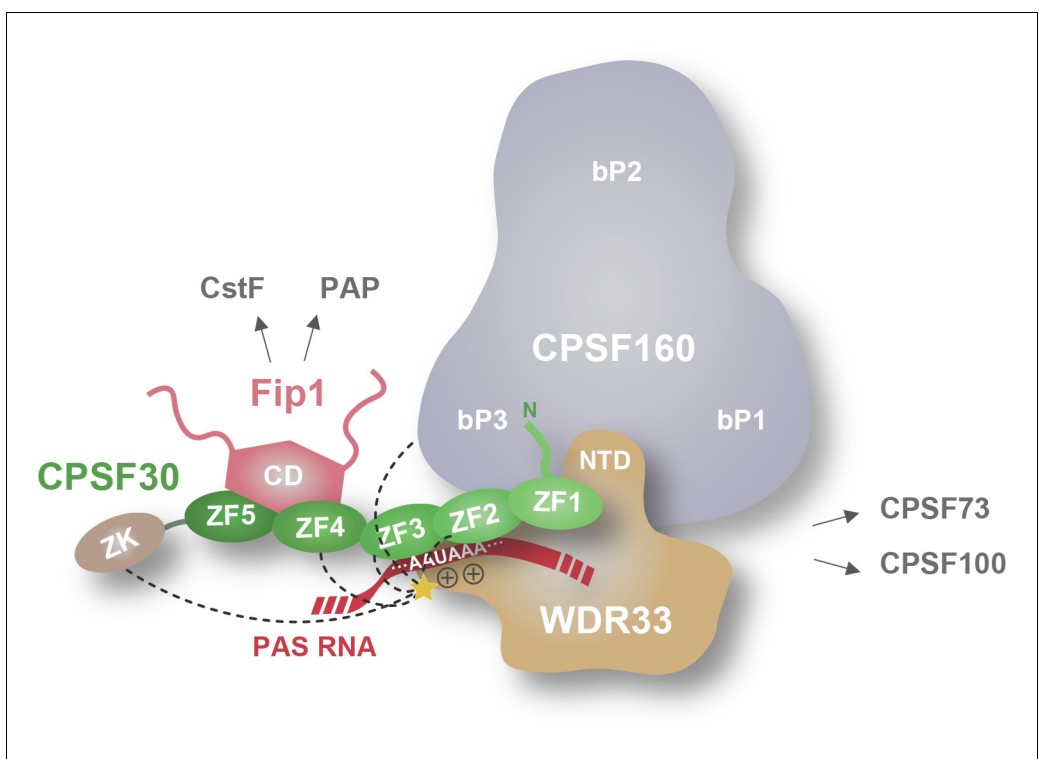

**Figure 5.** Schematic model of the mammalian core CPSF complex bound to the PAS hexamer motif. CPSF160 and WDR33 form heterodimer acting as a structural scaffold of the core CPSF complex. WDR33 interacts with CPSF160 through its NTD domain which is inserted in deep cavity formed by CPSF160 N- and C-terminal beta-propeller domains (bP1 and bP3). CPSF30 connects CPSF160-WDR33 to Fip1, interacting with the former via its N-terminus (including the zinc finger domain ZF1) and with the latter via its C-terminal zinc finger domains ZF4-5. The hexanucleotide AAUAAA sequence is bound by CPSF30 ZF2-3 and a positively charged lysine/arginine-rich motif located N-terminally of the NTD domain in WDR33. The latter is also a lysine cross-linking hot spot (indicated by a yellow star) with CPSF30 and CPSF160. The dark dashed lines indicate the CPSF30 and CPSF160 domains contacted by the cross-linking hotspot.

DOI: https://doi.org/10.7554/eLife.33111.015

preparation, a cryoEM structure of the human CPFS160-WDR33-CPSF30-Fip1 complex bound to the AAUAAA motif RNA was published, revealing that both the ZF2 and ZF3 domains of CPSF30 and the N-terminal region are responsible for sequence-specific recognition of the AAUAAA motif (*Sun et al., 2017*), in agreement with our structural and biochemical insights. Together, these studies establish the structural framework for polyadenylation signal recognition by the core CPSF complex and set the stage for further structural and mechanistic studies of the mammalian polyadenylation machinery.

## Materials and methods

### Protein expression and purification

All constructs of human CPSF160 (Uniprot Q10570), WDR33 (Uniprot Q9C0J8-1), CPSF30 (Uniprot O95639-3) and Fip1 (Uniprot Q6UN15-4) were cloned into ligation-independent MacroLab vectors developed by Scott Gradia, University of California, Berkeley (*Gradia et al., 2017*). The constructs were cloned in single-cassette, double-cassette or 438 MacroBac cloning system vectors as specified in *Supplementary file 1*. For 438 MacroBac cloning system vectors, two, three or four subunits, as appropriate, were combined in a single plasmid according to the MacroBac protocol (*Gradia et al., 2017*) as specified in *Supplementary file 1*. Recombinant baculoviruses were generated using the Bac-to-Bac system (Invitrogen) according to standard protocols. Sf9 insect cells were either infected with one virus or co-infected with two viruses as specified in *Supplementary file 1*. GST-Fip1$^{CD}$ and MBP-CPSF30 constructs used in pull-down experiments were cloned in the 2GT (Addgene #29707) and 1M (Addgene #29656) vectors, respectively.

Recombinant CPSF complexes were expressed in Sf9 cells infected at a density of $1.0 \times 10^6$ ml$^{-1}$. For purifications of the CPSF160$^{FL}$-WDR33$^{Q35-K410}$ complex used for crystallization, cells were harvested 72 hr post infection, resuspended in 20 mM Tris-Cl pH 7.5, 150 mM NaCl, and 0.05% Tween20, supplemented with Protease Inhibitor Cocktail (GE Healthcare), and lysed by sonication. The complex was purified on Ni-NTA resin (Qiagen GmbH, Hilden, Germany) followed by purification on glutathione sepharose 4 fast flow resin (GE Healthcare BioSciences AB, Uppsala, Sweden). After removal of the tags by digestion with TEV protease, the complex was further purified by size exclusion chromatography on a Superdex-200 column (GE Healthcare) in 20 mM HEPES pH 7.5, 150 mM KCl and 1 mM DTT.

For the purification of CPSF complexes used in fluorescence polarization RNA-binding measurements, cells were harvested 72 hr post infection, resuspended in 20 mM Tris-HCl pH 7.5, 300 mM NaCl, 10% glycerol, 0.05% Tween20, supplemented with Protease Inhibitor Cocktail (GE Healthcare), and lysed by sonication. For complexes containing Fip1, 200 mM NaCl was used. The complexes were purified on Ni-NTA resin (Qiagen) followed by purification on Streptactin superflow resin (IBA GmbH, Goettingen, Germany) and size exclusion chromatography on a Superdex-200 column (GE Healthcare) in 20 mM Tris-HCl pH 7.5, 150 mM NaCl and 1 mM DTT.

GST-Fip1$^{CD}$ and MBP-CPSF30 were expressed in *E. coli* BL21 Star (DE3) cells grown until an OD$_{600}$ of 0.6 and the expression was induced by addition of isopropyl-1-thio-β-D-galactopyranoside (IPTG) to a final concentration of 0.1 mM. GST-Fip1$^{CD}$ was expressed at 20°C overnight and MBP-CPSF30 at 37°C for 3 hr. Cells were harvested, resuspended in 20 mM Tris-HCl pH 7.5 and 500 mM NaCl supplemented with 1 mM PMSF/AEBSF protease inhibitor, and lysed by sonication. GST-Fip1$^{CD}$ was purified on glutathione sepharose 4 fast flow resin (GE Healthcare) and MBP-CPSF30 on amylose resin (NEB Inc., Ipswich, MA, USA). Both proteins were further purified by size exclusion chromatography on a Superdex-200 column (GE Healthcare) in 20 mM Tris-HCl pH 7.5, 150 mM NaCl and 1 mM DTT.

### Cross-linking and mass spectrometry analysis

80 µg of purified CPSF160$^{FL}$-WDR33$^{M1-G474}$-CPSF30$^{FL}$-Fip1$^{FL}$ complex was cross-linked at a concentration of 2 mg ml$^{-1}$ with a final concentration of 1 mM equimolar mixture of isotopically labeled disuccinimidyl suberate (DSS-d$_0$, DSS-d$_{12}$; CreativeMolecules Inc.) in a final volume of 40 µl of 20 mM HEPES pH 7.5, 150 mM NaCl, 0.25 mM TCEP, incubated at 37°C for 30 min at 500 rpm on a Thermomixer (Eppendorf AG, Hamburg, Germany), as previously described (*Leitner et al., 2014*). The reactions were quenched with 50 mM (final concentration) of ammonium bicarbonate (NH$_4$HCO$_3$) for

20 min at 37°C and evaporated to dryness in a vacuum centrifuge. The dried pellets were dissolved in 50 µl of 8 M urea, reduced with 2.5 mM TCEP for 30 min at 37°C and alkylated with 5 mM iodoacetamide (Sigma-Aldrich) for 30 min at room temperature, in the dark. Digestion was carried out after diluting urea to 5 M with 50 mM $NH_4HCO_3$ and adding 1% (w/w) LysC protease (Wako Chemicals GmbH, Neuss, Germany) for 3 hr at 37°C and subsequently diluting to 1 M urea with 50 mM $NH_4HCO_3$ and finally adding 2% (w/w) trypsin (Promega, Fitchburg, WI, USA) for 14 hr at 37°C. Protein digestion was stopped by acidification with 1% (v/v) formic acid. Digested peptides were purified using Sep-Pak C18 cartridges (Waters, Milford, MA, USA) according to the manufacturer's protocol. Cross-linked peptides were enriched by peptide size exclusion chromatography (SEC) as previously described (*Leitner et al., 2014*). SEC fractions were then reconstituted in 5% acetonitrile and 0.1% formic acid and analyzed in duplicates on an HPLC (Thermo Easy-nLC 1000) coupled to a mass spectrometer (Thermo Orbitrap Elite). Analytes were separated on an Acclaim PepMap RSLC column (25 cm x 75 µm, 2 µm particle size, Thermo Scientific, Waltham, MA, USA) over a 60 min gradient from 7% to 35% acetonitrile at a flow rate of 300 nl $min^{-1}$. The mass spectrometer was operated in data-dependent acquisition (DDA) mode with MS acquisition in the Orbitrap analyzer at 120,000 resolution and MS/MS acquisition in the linear ion trap at normal resolution after collision-induced dissociation. DDA was set up to isolate the top 10 precursors from an MS1 full scan with a charge state of +3 or higher and a dynamic exclusion of 30 s (*Leitner et al., 2014*). MS data were converted to mzXML format with msConvert (*Chambers et al., 2012*) and searched with xQuest/xProphet (*Walzthoeni et al., 2012*) with an MS1 tolerance of 10 ppm and an MS2 tolerance of 0.2 and 0.3 Da for common ions and cross-link ions, respectively. Default settings for xQuest and xProphet settings were selected (*Leitner et al., 2014*). The search database contained the peptide sequences of the analyzed proteins and the corresponding decoy sequences. Cross-linked peptides were identified with a minimal length of 5 amino acids and at least four bond cleavages or three adjacent ones per peptide. Cross-linked peptides were defined as validated if they had a total ion current explained higher than 0.1 and an xQuest score higher than 30, corresponding to an approximate false discovery rate of 10%. 18 (12%) of the 153 validated cross-linked sites could be mapped to residues present in the atomic model of CPSF160-WDR33. From the subset of mapped cross-linked residues, 12 had an Euclidean distance lower that 35 Å, whereas six showed a distance between 42 and 100 Å. Notably, 5 of the six cross-linked sites originated from a common peptide on CPSF160 (682-LALHKPPLHHQSK-694). Figures were prepared with xiNET (*Combe et al., 2015*), UCSF Chimera (*Pettersen et al., 2004*) and UCSF Chimera X (*Goddard et al., 2018*).

## CPSF160-WDR33 complex crystallization and structure determination

Crystals of $CPSF160^{FL}$-$WDR33^{Q35-K410}$ were obtained using the hanging drop vapor diffusion method by mixing 0.5 µl of protein at 4.4 mg/ml and 0.5 µl of reservoir solution containing 11% (w/v) PEG 3400, 37.5 mM ammonium formate and 112.5 mM magnesium formate (native data set) or 10% w/v PEG 3400 and 140 mM magnesium formate (Sulfur SAD data set). For derivatization with tantalum bromide ($Ta_6Br_{12}$), crystals were grown in 8% w/v PEG3400 and 70 mM magnesium formate and soaked in the reservoir solution containing 1 mM $Ta_6Br_{12}$ for one hour. For data collection, crystals were cryoprotected by transfer to a solution containing 20% (v/v) glycerol and 80% (v/v) of reservoir solution and flash cooled in liquid nitrogen. X-ray diffraction data were collected at beam line X06DA (PXIII) of the Swiss Light Source (Paul Scherrer Institute, Villigen, Switzerland) and processed using XDS (*Kabsch, 2010*). Data collection statistics are shown in *Table 1*. Native and $Ta_6Br_{12}$-SAD data comprised four data sets and native sulfur-SAD comprised 17 data sets collected on two different crystals. All data sets were collected by exposing different parts of the same crystal, rotating the crystal through 360° in each data set and changing the kappa angle between datasets in 5° increments to ensure data completeness and redundancy. The structure was solved by a combination of molecular replacement (MR) and single-wavelength anomalous diffraction (SAD) using the Phaser module in Phenix (*Adams et al., 2010*). A low-score MR solution (TFZ = 4) was obtained for the native data set using a polyalanine model based on the DDB1 structure (PDB entry 3EI3) as search model. The solution was subsequently used to perform MR-SAD on the sulfur- and $Ta_6Br_{12}$-SAD datasets. A homology model of CPSF160 was fitted in the map obtained by $Ta_6Br_{12}$ MR-SAD, manually rebuilt using Coot (*Emsley and Cowtan, 2004*) with the aid of the sulfur site positions identified by sulfur-MR-SAD and refined using phenix.refine (*Afonine et al., 2012*). The resulting atomic model was used to perform MR-SAD on the sulfur-SAD dataset, which yielded an improved

map and additional sulfur sites. After several iterations of MR-SAD and manual building, whereby the manually optimized model was used as input for sulfur MR-SAD yielding an optimized map that allowed further building of the model, a homology model of WDR33 beta-propeller domain could be fitted in the map and manually refined. Finally, the CPSF160-WDR33 model was used to perform molecular replacement in the high resolution native data and manually built and improved using Coot and refined using phenix.refine.

## Pull-down assays

For pull-down assays from Sf9 cells expressing CPSF subunits, the cells were resuspended in 20 mM Tris-Cl pH 7.5, 200 mM NaCl, 10% glycerol and 0.05% Tween20, and lysed by sonication. The clarified lysate was incubated with 600 µl of NiNTA resin (Qiagen). After washing, the protein was eluted and incubated directly with 30 µl Streptactin sepharose beads (GE Healthcare). The beads washed three times with 1 ml of resuspension buffer and the protein eluted with SDS-PAGE loading buffer at room temperature. The eluted samples and the input lysate were loaded on SDS-PAGE with no prior boiling to avoid GFP denaturation, visualized on a Typhoon FLA9500 fluorescence scanner (GE Healthcare) and subsequently stained with Coomassie brilliant blue R250.

For pull-down assays with purified proteins, 3 µg of GST-Fip1$^{CD}$ was immobilized on glutathione sepharose four fast flow beads (GE Healthcare) and incubated with 40 µg of MBP-CPSF30. The beads were washed three times with 500 µl of buffer containing 20 mM Tris-Cl pH 7.5, 150 mM NaCl and 0.05% Tween20, the sample eluted with SDS-PAGE loading buffer and analyzed by SDS-PAGE.

## Fluorescence polarization RNA binding assays

Equilibrium binding experiments were carried out in a PheraStar FSX fluorescence plate reader (BMG Labtech, Ortenberg, Germany) at 25°C in 20 mM Tris pH 7.5, 150 mM NaCl, 1 mM DTT and 0.05% Tween 20 in 384-well non-binding surface, flat bottom black plates (Greiner) and 50 µl final volume. The CPSF complex at different concentrations was incubated with 0.1 nM 5'-Atto$^{532}$-labelled L3 PAS RNA, either wild-type (CACACAAUAAAGGCAA) or mutant (CACACAAGAAAGG-CAA). Fluorescence polarization was measured with an excitation filter with a central wavelength of 540 nm, and P and S emission filters with a central wavelength of 590 nm. The FP values were plotted as a function of concentration and fitted to a one-site binding model accounting for ligand depletion using Prism6 (GraphPad, Inc.). The polarization amplitude was normalized to 1. Error bars indicate standard error of means (SEM) for five consecutive measurements of a single representative sample. Each binding experiment was performed in triplicate and the mean $K_D$ values ± SEM are reported in *Table 2*.

## Acknowledgements

We are grateful to Meitian Wang, Vincent Olieric and Takashi Tomizaki at the Swiss Light Source (Paul Scherrer Institute, Villigen, Switzerland) for assistance with x-ray diffraction measurements. We thank members of the Jinek group for discussions and critical reading of the manuscript. This work was supported by the European Research Council (ERC) Starting Grant ANTIVIRNA (Grant no. ERC-StG-337284). MF was supported by a long-term fellowship from the European Molecular Biology Organization (EMBO ALTF-343–2013). RA acknowledges support from the European Union 7th Framework Program (PROSPECTS, HEALTH-F4-2008-201648), the European Research Council (ERC Advanced Grants no. 233226 and no. 670821), and the Innovative Medicines Initiative Joint Undertaking (ULTRA-DD, grant no. 115766). MJ is International Research Scholar of the Howard Hughes Medical Institute and Vallee Scholar of the Bert L and N Kuggie Vallee Foundation.

## Additional information

### Funding

| Funder | Grant reference number | Author |
| --- | --- | --- |
| European Research Council | ERC-StG-337284 | Marcello Clerici<br>Martin Jinek |

| European Molecular Biology Organization | ALTF-343-2013 | Marco Faini |
|---|---|---|
| European Research Council | HEALTH-F4-2008-201648 | Ruedi Aebersold |
| European Research Council | 233226 | Ruedi Aebersold |
| H2020 European Research Council | 670821 | Ruedi Aebersold |
| Innovative Medicines Initiative Joint Undertaking | ULTRA-DD grant no. 115766 | Ruedi Aebersold |
| Howard Hughes Medical Institute | | Martin Jinek |

The funders had no role in study design, data collection and interpretation, or the decision to submit the work for publication.

### Author contributions

Marcello Clerici, Conceptualization, Data curation, Formal analysis, Validation, Investigation, Writing—original draft, Writing—review and editing, Designed experiments, Prepared all samples, determined the crystal structure of CPSF160-WDR33 and carried out biochemical assays; Marco Faini, Validation, Investigation, Writing—review and editing, Performed cross linking and mass spectrometry analysis under the supervision of RA; Ruedi Aebersold, Supervision, Methodology, Writing—review and editing; Martin Jinek, Conceptualization, Supervision, Funding acquisition, Investigation, Writing—original draft, Project administration, Writing—review and editing, Designed the experiments, Assisted with x-ray structure determination

### Author ORCIDs

Marcello Clerici (iD) http://orcid.org/0000-0003-2906-0982
Marco Faini (iD) http://orcid.org/0000-0001-8131-7648
Martin Jinek (iD) http://orcid.org/0000-0002-7601-210X

### Decision letter and Author response

Decision letter https://doi.org/10.7554/eLife.33111.024
Author response https://doi.org/10.7554/eLife.33111.025

# Additional files

### Supplementary files

• Supplementary file 1. The table contains a list of the protein constructs expressed in Sf9 insect cells, their respective expression tags and the vector in which the constructs were cloned. The table also specifies which protein constructs were combined in a single vector using the MacroBac system (see Materials and methods), and whether a single baculovirus was used for protein expression or a combination of two baculoviruses.
DOI: https://doi.org/10.7554/eLife.33111.016

• Transparent reporting form
DOI: https://doi.org/10.7554/eLife.33111.017

### Major datasets

The following datasets were generated:

| Author(s) | Year | Dataset title | Dataset URL | Database, license, and accessibility information |
|---|---|---|---|---|
| Clerici M, Jinek M | 2017 | Crystal structure of the human CPSF160-WDR33 complex | http://www.rcsb.org/pdb/search/structid-Search.do?structureId=6F9N | Publicly available at the RCSB Protein Data Bank (accession no. 6F9N) |

| Faini M, Aebersold R | 2017 | CPSF160-WDR33-CPSF30-Fip1 MS raw data and cross-linking results | http://proteomecentral.proteomexchange.org/cgi/GetDataset?ID=PXD008122 | Publicly available at ProteomeXchange (accession no. PXD008122) |

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
