## [Decision Letter]

Thank you for submitting your article "Structural insights into the assembly and polyA signal recognition mechanism of the human CPSF complex" for consideration by *eLife*. Your article has been reviewed by three peer reviewers, one of whom, Nick J Proudfoot (Reviewer #1), is a member of our Board of Reviewing Editors, and the evaluation has been overseen by Kevin Struhl as the Senior Editor.

The reviewers have discussed the reviews with one another and the Reviewing Editor has drafted this decision to help you prepare a revised submission.

This study presents the first high resolution x-ray crystallographic structure for the central RNA recognition subunits of the cleavage polyadenylation complex in mammals, the heterodimer CPSF160-WDR33 (N-terminal ~ 400aa). It also investigates protein-protein interactions within the reconstituted 4 subunit complex consisting of CPSF160, N-terminal part of WRD33, CPSF30 and FIP1 using cross-linking coupled to mass spectrometry and pull-down assays. Insight is obtained as to how this module achieves selective interaction with the PAS, substantially extending the original papers that demonstrated the role of WDR33 in cleavage and polyadenylation (Chan et al., 2014 and Schonemann et al., 2014). All three reviewers find this study to report interesting findings that nicely complement the recent study from the Passmore lab on the polyadenylation module from yeast (Casanal et al., 2017). In view of the competitive nature of this research we do not suggest any further experimental analysis, but do recommend a fair number of changes and clarifications to the manuscript that we feel are necessary for it to be ready for *eLife* publication.

1) Casanal et al. should be referenced and discussed in context of the reported data.

2) The PAS is not just the AAUAAA sequence but is rather a combination of USE-AAUAAA-DSE. AAUAAA should be described as the central hexamer sequence of the PAS but not the PAS itself. Original references on PAS sequence should be cited.

3) Domains and their positions (aa) should be clearly indicated on schematics in Figure 1 and domain boundaries should be made clearer. For example, it is unclear if the coloured part of WDR33 comprises just the β propeller domain or includes the structured N-terminus. This is relevant when judging where the cross-link sites are as discussed extensively in the paper. Abbreviations (such as 'CD' in FIP1, CTD in CPSF160, ZK in CPSF30) should be explained in figure legend.

4) Figure 2: It would be helpful to colour-code the three domains of CPSF160. Figure 2—figure supplement 1 – selection of the colours makes them are hard to distinguish and precludes appreciation of the structural similarity.

5) An explanation of why RNA binding is increased when truncated CPSF30 is used (Figure 4) is required. Does the absence of Fip1 contribute to this?

6) MS analyses and interpretation. In subsection “Crystal structure of the CPSF160-WDR33 heterodimer reveals the structural scaffold of the CPSF complex”, it should be more explicitly stated how many X-links were or were not consistent with their structure. In subsection “Molecular topology of the core CPSF complex”, final paragraph, the sentence 'These conclusions are consistent […]' should be rephrased as the argument made is unclear. Fip1 is mentioned in the beginning of the sentence, but data concerning this subunit are not mentioned. The corresponding Figure 3—figure supplement 2 shows the sites of cross-linking on WDR33 and CPSF160. It would be useful to indicate more precisely which parts of Fip1 and CPSF30 are cross-linked to these sites?

7) The cross-linking hot spot at the N-terminus of WDR33 is discussed quite a bit. It appears that the cross-linked amino acids are very close to the part of WDR33 visible in the x-ray structure. In addition, the very many cross-links of this short amino acid sequence, including some to other parts of the known structure, provide many constraints. It would be useful to present this information in a model. Figure 5 is helpful, but more details are still necessary. Mass spectrometric raw data as well as xQuest/xProphet output and setting files should be made publicly available via an appropriate data repository. What are 'validated' cross-linked sites?

8) Figure 3 and text in subsection “Molecular topology of the core CPSF complex”: It is unclear how the pull-down assays were performed. In panel A, the his-strep tag on WDR33 was presumably used, but it is hard to see which part of the tag was used. In panel C, lane 6, WDR33 was not present, so it is unclear how the precipitation was done. Asterisks in panel A and B are not explained.

Figure 3, Figure 3—figure supplement 1, panel B shows a re-analysis of Figure 3. Why not simply replace Figure 3?

9) Discussion: The first paragraph partially repeats the Introduction and could be omitted. Cite Schonemann et al. for the first sentence, second paragraph.

Here they reconstituted the four subunit complex from co-purified WDR33 – Fip1 and CPSF160 – CPSF30 heterodimers. The authors might wish to discuss how this fits with the interactions they see. For example, reconstitution from WDR33 and CPSF160 expressed separately from each other is inconsistent with the authors' idea that the interaction between the two might require co-folding durch assembly.

Mammalian CstF and yeast CF I are not exactly equivalent. The authors might state that one is related to the other.

10) Materials and methods:

The authors might provide accession numbers as the basis for the amino acid residue numbers given.

Please provide a reference for the 'MacroLab' vectors (I found a paper in Methods in Enzymology).

---

## [Author Response]

1) Casanal et al. should be referenced and discussed in context of the reported data.

In light of the recent publications from the Passmore and Tong groups, we have rewritten the Discussion section of the manuscript to set our structural and biochemical insights in the context of these studies. We first note that the yeast Cft1-Pfs2-Yth1-Fip1 complex structure is similar to the crystal structure of the human CPSF160-WDR33 heterodimer.

“A recent cryo-EM structure of the yeast Cft1-Pfs2-Yth1-Fip1 complex, which is orthologous and functionally equivalent to the human CPSF160-WDR33-CPSF30-Fip1 complex and constitutes the polyadenylation module of the yeast CPF complex, revealed striking similarities to the crystal structure of the CPSF160-WDR33 heterodimer (Casanalet al., 2017). The yeast Cft1-Pfs2 subcomplex superimposes with the human CPSF160-WDR33 heterodimer with RMSDs of 2.56 Å for CPSF1160 and Cft1 and 1.55 Å for WDR33 and Pfs2, indicating that the core CPSF scaffold is highly evolutionarily conserved.”

We also make further reference to the Casanal et al. study in the subsequent paragraph and state: “Our interaction data is consistent with the yeast Cft1-Pfs2-Yth1-Fip1 complex structure, in which the N-terminal region of Yth1 up to and including ZF1 mediates the interaction with Cft1-Pfs2 (CPSF160WDR33), while the C-terminal ZF4 and ZF5 domains, together with Fip1, are structurally disordered (Casanal et al., 2017). Notably, Yth1 ZF1 and ZF2 domains are located in the cleft formed by Cft1 and Pfs2, in good agreement with our cross-linking and mass-spectrometry data.”

In the final paragraph of the Discussion section we now cite the Sun et al. PNAS paper and state:

“While this manuscript was in preparation, a cryoEM structure of the human CPFS160-WDR33CPSF30-Fip1 complex bound to the AAUAAA motif RNA was published, revealing that both the ZF2 and ZF3 domains of CPSF30 and the N-terminal region are responsible for sequence-specific recognition of the AAUAAA motif (Sun et al., 2017), in agreement with our structural and biochemical insights. Together, these studies establish the structural framework for polyadenylation signal recognition by the core CPSF complex and set the stage for further structural and mechanistic studies of the mammalian polyadenylation machinery.”

2) The PAS is not just the AAUAAA sequence but is rather a combination of USE-AAUAAA-DSE. AAUAAA should be described as the central hexamer sequence of the PAS but not the PAS itself. Original references on PAS sequence should be cited.

We apologize for the inconsistency in our use of the PAS nomenclature. We have changed the term “PAS RNA” (when referring to the AAUAAA motif) to “PAS hexamer motif” or “AAUAAA motif” throughout the text.

We have also added the following original references for the distinct cis-elements within the polyadenylation signal:

AAUAAA:

Proudfoot NJ and Brownlee GG (1976). "3' non-coding region sequences in eukaryotic messenger RNA." Nature 263(5574): 211-214.

USE:

Brackenridge S and Proudfoot NJ (2000). "Recruitment of a basal polyadenylation factor by the upstream sequence element of the human lamin B2 polyadenylation signal." Mol Cell Biol 20(8): 26602669.

Carswell S and Alwine JC (1989). "Efficiency of utilization of the simian virus 40 late polyadenylation site: effects of upstream sequences." Mol Cell Biol 9(10): 4248-4258.

DSE:

Gil A and Proudfoot NJ (1984). "A sequence downstream of AAUAAA is required for rabbit β-globin mRNA 3'-end formation." Nature 312(5993): 473-474.

Gil A and Proudfoot NJ (1987). "Position-dependent sequence elements downstream of AAUAAA are required for efficient rabbit β-globin mRNA 3' end formation." Cell 49(3): 399-406.

McLauchlan J, Gaffney D, Whitton JL and Clements JB (1985). "The consensus sequence YGTGTTYY located downstream from the AATAAA signal is required for efficient formation of mRNA 3' termini." Nucleic Acids Res 13(4): 1347-1368.

3) Domains and their positions (aa) should be clearly indicated on schematics in Figure 1 and domain boundaries should be made clearer. For example, it is unclear if the coloured part of WDR33 comprises just the β propeller domain or includes the structured N-terminus. This is relevant when judging where the cross-link sites are as discussed extensively in the paper. Abbreviations (such as 'CD' in FIP1, CTD in CPSF160, ZK in CPSF30) should be explained in figure legend.

We apologize for the unclear schematics in Figure 1 and we agree that this is relevant to map the cross-linking sites on the protein. In order to clarify this point, we have indicated domain boundaries (residue number) in Figure 1 and Figure 1—figure supplement 1 and added the WDR33 N-terminal domain (NTD) to the schematic representation of WDR33 (with the relative domain boundaries). Abbreviations are explained in the figure legends for Figure 1 and Figure 1—figure supplement 1 (CD, Fip1 conserved domain; ZF1-5, CPSF30 zinc finger domains 1 to 5; ZK, CPSF30 zinc knuckle domain; CTD, CPSF160 C-terminal domain; bP1-3, CPSF160 β-propeller domains 1 to 3; NTD, WDR33 Nterminal domain).

4) Figure 2: It would be helpful to colour-code the three domains of CPSF160. Figure 2—figure supplement 1 – selection of the colours makes them are hard to distinguish and precludes appreciation of the structural similarity.

We thank the Reviewer for the suggestion. We have color-coded the three propellers and the C-terminal domains of CPSF160 in Figure 2 and reported the corresponding colors in CPSF160 schematic representation in Figure 1 and Figure 1—figure supplement 1. In Figure 2—figure supplement 1 we have changed the color of DDB1 and DDB2 to increase the contrast with respect to CPSF160 and WDR33 to facilitate the comparison between the two proteins.

5) An explanation of why RNA binding is increased when truncated CPSF30 is used (Figure 4) is required. Does the absence of Fip1 contribute to this?

The observed reduced affinity of the CPSF complex carrying CPSF30 ZF4-5 and Fip1 for the AAUAAA RNA motif is presently not clear. A possible explanation could be that these domains sterically hinder the access of the AAUAAA RNA motif to CPSF30 ZF2-3. We added the following sentence in the Results section describing the CPSF-AAUAAA motif affinity measurements: “Although the observed increase in affinity upon removal of CPSF30 domains ZF4-5 and Fip1 is presently unclear, it is conceivable that CPSF30 ZF4-5 and/or Fip1 play an autoinhibitory role within CPSF by sterically hindering the accessibility of CPSF30 ZF2-3 domains to the AAUAAA motif.”

6) MS analyses and interpretation. In subsection “Crystal structure of the CPSF160-WDR33 heterodimer reveals the structural scaffold of the CPSF complex”, it should be more explicitly stated how many X-links were or were not consistent with their structure.

We have added a sentence in the text, stating the number of cross-linked sites compatible with the conventional upper limit Euclidean distance expected by DSS cross-linking (~ 35 Å). We have also added an estimate of the false discovery rate and detailed the origin and extent of the distance violations in the cross-linking mass spectrometry analysis section (Results section second paragraph). The subset of mapped cross-links that are more than 35 Å apart have proportionally lower (poorer) xQuest ld-scores (see Author response image 1) and therefore are expected to be less reliable. The number of distance-violating cross-links (6) is within the expected false discovery rate of 10% (15 expected false cross-links). We also note that many of the cross-links originated from lysine residues within disordered regions of CPSF160 (internal loops) and WDR33 (N-terminal region), therefore limiting our ability to verify whether the conventional Euclidean distance of 35 Å was violated in their case. We also added this information in the Results section.

**Author response image 1. respfig1:** ld-score distribution of cross-link sites. The 153 cross-linking sites are depicted in green bins whereas the cross-links that are violating the distance threshold of 35 Å are colored in red bins. The cross-links that are violating the distance threshold have generally poorer scores.

In subsection “Molecular topology of the core CPSF complex”, final paragraph, the sentence 'These conclusions are consistent […]' should be rephrased as the argument made is unclear. Fip1 is mentioned in the beginning of the sentence, but data concerning this subunit are not mentioned.

We have now specified that the CPSF complex topology inferred from pull down data is consistent with the cross-linking data between each subunit presented in Figure 1 (subsection “AAUAAA motif recognition is mediated by CPSF30 ZF2-3 domains and an N-terminal motif in WDR33”). We state”Furthermore, these conclusions are consistent with the cross-linking data between each subunit (Figure 1), as mapping the molecular cross-links of CPSF30 and Fip1 onto the molecular surface of the CPSF160-WDR33 heterodimer suggests that CPSF30 binds at, or close to, the CPSF160-WDR33 molecular interface in proximity to the N-terminus of WDR33 (Figure 3—figure supplement 2).

The corresponding Figure 3—figure supplement 2 shows the sites of cross-linking on WDR33 and CPSF160. It would be useful to indicate more precisely which parts of Fip1 and CPSF30 are cross-linked to these sites?

As suggested by the Reviewers, we have added two sequence graphs for CPSF30 and Fip1 to Figure 3—figure supplement 2 and mapped the cross-linked sites onto the surface of the CPSF160-WDR33 complex to the corresponding sites on the respective sequences. WDR33 K50 was cross-linked to several lysine residues in CPSF30 ZF2-4 domains but disordered in the crystal structure. Since K50 is in close proximity to the last ordered residue of the crystal structure (R54), we have depicted a line representing WDR33 N-terminal extension carrying the K50 residue, in order to have a more complete overview of the cross-links between WDR33 N-terminal cross-linking hotspot and CPSF30.

7) The cross-linking hot spot at the N-terminus of WDR33 is discussed quite a bit. It appears that the cross-linked amino acids are very close to the part of WDR33 visible in the x-ray structure. In addition, the very many cross-links of this short amino acid sequence, including some to other parts of the known structure, provide many constraints. It would be useful to present this information in a model. Figure 5 is helpful, but more details are still necessary.

We believe that the most relevant information yielded by the cross-linking mass spectrometry experiments is the proximity of the cleft formed by CPSF160-WDR33 (including WDR33 N-terminal “hotspot”) and CPSF30 zinc finger domains (in agreement with Casanal et al., 2017) and the middle portion of Fip1 including its conserved domain (CD). We summarize this information in the updated version of Figure 3—figure supplement 2 and in Figure 5. We decided to exclude from the schematic model the cross-links between CPSF160 and WDR33 which map on the atomic coordinates since the information implicit in the crystal structure is more accurate and, therefore, the cross-link data, although in good general agreement, is redundant in this context. Additionally, as explained in our answer to point (6), many cross-links map on disordered regions of the crystal structure, restricting the amount of cross-links that can be correctly displayed.

Mass spectrometric raw data as well as xQuest/xProphet output and setting files should be made publicly available via an appropriate data repository. What are 'validated' cross-linked sites?

We apologize that the use of the term validated was not sufficiently clear from the text. We have now added the sentence:” (xQuest score higher than 30, corresponding to an approximate false discovery rate of 10%)” after “validated” in the second paragraph of the Results section. We have also clarified in the Materials and methods section the meaning of validated cross-links by explicitly defining them. We have also added an estimate of the false discovery rate. Furthermore, we have specified the search mass tolerances and a reference for the default xQuest and xProphet settings (Leitner et al., 2014). These settings are also available in the data processing protocol section of the PRIDE archive submission (see below). We have made the mass spectrometric raw files and the search results publicly available from the PRIDE archive through ProteomeXchange with the accession number: PXD008122.

We have added the sentence: “The MS raw data and the cross-linking results are available via ProteomeXchange with identifier PXD008122. “to the Additional Files section.

8) Figure 3 and text in subsection “Molecular topology of the core CPSF complex”: It is unclear how the pull-down assays were performed. In panel A, the his-strep tag on WDR33 was presumably used, but it is hard to see which part of the tag was used.

The pull-down was a tandem affinity purification performed initially on nickel-affinity beads and then incubating the eluted material with Streptactin beads, as detailed in the Materials and methods section. This tandem purification ensures the purity of the eluted material and a more stringent selection of the precipitated proteins, avoiding spurious interactions. For clarity, we explicitly mention the use of tandem affinity purification: “In tandem affinity purifications utilizing the His_6_-(StrepII)_2_ epitope tag on WDR33, we initially observed […]”.

In panel C, lane 6, WDR33 was not present, so it is unclear how the precipitation was done.

GFP-Fip1^CD^ was co-expressed with His_6_-(StrepII)_2_-CPSF30^ZF4-5^ and the latter used as bait for the pulldown in the absence of CPSF160^FL^-WDR33^M1-K410^ to verify that the isolated CPSF30^ZF4-5^ is sufficient to mediate the interaction with Fip1^CD^. We changed the sentence to explicitly state that CPSF30 is His_6_-(StrepII)_2_-tagged:”In the absence of CPSF160 and WDR33, a His_6_-(StrepII)_2_–tagged fragment of CPSF30 comprising the ZF4 and ZF5 domains (CPSF30^ZF4-5^) was able to interact with Fip1^CD^ (Figure 3, lane 6)”.

For clarity, we added an additional explanation in the legend of Figure 3: “In lane 6, His_6_-(StrepII)_2_-CPSF30^ZF4-5^ was used for the precipitation of GFP-Fip1^CD^, in the absence of WDR33^M1-K410^ and CPSF160^FL^”

Asterisks in panel A and B are not explained.

The sentence: “The asterisk indicates free GFP present in the input lysate.” was added in the legend of Figure 3 and B.

Figure 3, Figure 3—figure supplement 1, panel B shows a re-analysis of Figure 3. Why not simply replace Figure 3?

Following the reviewer’s suggestion, we moved panel B of Figure 3—figure supplement 3 as panel B of Figure 3.

9) Discussion: The first paragraph partially repeats the Introduction and could be omitted. Cite Schonemann et al. for the first sentence, second paragraph.Here they reconstituted the four subunit complex from co-purified WDR33 – Fip1 and CPSF160 – CPSF30 heterodimers. The authors might wish to discuss how this fits with the interactions they see. For example, reconstitution from WDR33 and CPSF160 expressed separately from each other is inconsistent with the authors' idea that the interaction between the two might require co-folding durch assembly.

Following the Reviewer’s suggestion, we removed the first paragraph of the Discussion section and added the reference at the end of the indicated sentence.

We thank the Reviewer for pointing out that the CPSF160-WDR33 complex forms also when the two proteins are expressed separately. We have removed the sentence “It is likely that this interaction is achieved by WDR33 and CPSF160 co-folding during CPSF assembly. These insights suggest that the CPSF160-WDR33 is an obligate heterodimer […]”.

Mammalian CstF and yeast CF I are not exactly equivalent. The authors might state that one is related to the other.

We thank the Reviewer for suggesting this clarification. We have corrected the relevant sentence as follows: “[…] which enables the cleavage and polyadenylation complexes to recruit, via Fip1, other polyadenylation factors including polyA polymerase and CStF (or the related factor CFI in yeast).”

10) Materials and methods:The authors might provide accession numbers as the basis for the amino acid residue numbers given.

We have added Uniprot accession numbers in the Materials and methods section and in the legend for Figure 1.

Please provide a reference for the 'MacroLab' vectors (I found a paper in Methods in Enzymology).

The MacroLab reference suggested by the Reviewer was already cited in the Materials and methods section. We have now added an additional citation at the beginning of the “Protein expression and purification” section:

“All constructs of human CPSF160 (Uniprot Q10570), WDR33 (Uniprot Q9C0J8-1), CPSF30 (Uniprot O95639-3) and Fip1 (Uniprot Q6UN15-4) were cloned into ligationindependent MacroLab vectors developed by Scott Gradia, University of California, Berkeley (Gradiaet al., 2017).”

For clarity, we have also added a table (Supplementary file 1) summarizing all the different constructs used and the vectors in which they were cloned (and their respective Addgene accession numbers), and shortened the first part of the “Protein expression and purification” section of Materials and methods.